# A Systematic Review of Commercial Smart Gloves: Current Status and Applications

**DOI:** 10.3390/s21082667

**Published:** 2021-04-10

**Authors:** Manuel Caeiro-Rodríguez, Iván Otero-González, Fernando A. Mikic-Fonte, Martín Llamas-Nistal

**Affiliations:** atlanTTic Research Center for Telecommunication Technologies, Universidade de Vigo, 36312 Vigo, Spain; iotero@gist.uvigo.es (I.O.-G.); mikic@gist.uvigo.es (F.A.M.-F.); martin@gist.uvigo.es (M.L.-N.)

**Keywords:** smart gloves, hand and finger pose estimation and motion tracking, haptic feedback, kinesthetic feedback, tactile feedback, extended reality

## Abstract

Smart gloves have been under development during the last 40 years to support human-computer interaction based on hand and finger movement. Despite the many devoted efforts and the multiple advances in related areas, these devices have not become mainstream yet. Nevertheless, during recent years, new devices with improved features have appeared, being used for research purposes too. This paper provides a review of current commercial smart gloves focusing on three main capabilities: (i) hand and finger pose estimation and motion tracking, (ii) kinesthetic feedback, and (iii) tactile feedback. For the first capability, a detailed reference model of the hand and finger basic movements (known as degrees of freedom) is proposed. Based on the PRISMA guidelines for systematic reviews for the period 2015–2021, 24 commercial smart gloves have been identified, while many others have been discarded because they did not meet the inclusion criteria: currently active commercial and fully portable smart gloves providing some of the three main capabilities for the whole hand. The paper reviews the technologies involved, main applications and it discusses about the current state of development. Reference models to support end users and researchers comparing and selecting the most appropriate devices are identified as a key need.

## 1. Introduction

Over the recent years, virtual, augmented and mixed reality systems (also known as extended reality or XR) have evolved significantly yielding enriched immersive experiences. Current low-cost head mounted displays (HMDs), such as the Oculus Rift or HTC Vive, provide high-fidelity 3D graphically-rendered environments that enable users to immerse in virtual experiences as never before. These solutions are specially focused on the visual and auditory senses. Nevertheless, for a more realistic experience, other senses should be considered, particularly haptic feedback based on kinesthetic and tactile interactions [1]. Research has already shown that users feel more immersed in XR if they can touch and get feelings in the forms of haptic interaction [2]. Similarly, interaction based on active movements contributes to the “*sense of agency*”, that is, the sense of having “*global motor control, including the subjective experience of action, intention, control, motor selection and the conscious experience of will*” [3].

To date, most commercial solutions use hand-based controllers with click buttons and inertial sensors for user interaction with XR devices. Even in many cases, vibration motors are included to provide some kind of haptic feedback. For example, when a collision with an object or a structure (e.g., a wall), a vibration alert is provided [4]. There are also solutions that perform some kind of body tracking, enabling to represent the user or a part of him/her (e.g., his/her hand) in the virtual scenario [5]. Nevertheless, these solutions are not perceived as natural [6], particularly because while users hold the provided controllers they cannot grab or touch objects in the virtual experience. Therefore, the development of real immersive XR demands other kinds of devices that facilitate a more natural human interaction, particularly freeing the user’s hands, recognizing gestures, and offering haptic feedback that allow users to feel what is happening in the virtual experience as if it was real [7].

From all the solutions considered, the concept of smart gloves is the most promising one in order to improve the immersive sensation, the degree of embodiment and presence in virtual/mixed reality [8,9,10]. Smart gloves are intended to enable users to touch and manipulate virtual objects in a more intuitive and direct way. They also pretend to provide sensitive stimuli that can be perceived by the human hands, particularly, kinesthetic and tactile feedback that simulates touching and manipulating objects. Non-functional requirements are also important: the glove device should be small, light, easy to carry, comfortable and it should not impair the motion and actions of the wearer. In addition, it should be adjustable to the variety of sizes and forms of human hands and fingers. There is a general understanding that this kind of device would enable users to experience more realistic XR, support patients’ rehabilitation, remote teleoperation, virtual surgery and experimentation, implementation of work sites, playing videogames, etc.

The vision for this kind of more interactive and immersive glove technology is nothing new. The first proposal of a hand-based device was done more than 40 years ago, in 1978 [11]. In 1982, Zimmerman applied for a patent (USA Patent 4542291) of a flexible optic sensor worn in a glove to measure the flexion of the fingers [12]. Zimmerman worked with Lanier to include ultrasonic and magnetic technology to track the hand position and create the Power Glove and the Data Glove (US Patent 4988981) [13]. Since then and along all these years, the pursuit of a device that facilitates hand-based interaction has been continuous, exploring different technologies and approaches. It is interesting to notice the different names used to refer to this kind of device (in alphabetical order): “cyber gloves“, “data gloves”, “force-feedback globes”, “glove-based systems“, “haptic gloves”, “sensory gloves”, “smart gloves”, “virtual gloves”, or “VR gloves”. Generally, the name is used to highlight some main purpose or device capability. For example, data gloves, by far the most frequent used name, highlights the capability to capture data from glove sensors, mainly related to hand and finger pose estimation and motion tracking. Meanwhile, haptic gloves are used to name those devices capable of providing some kind of kinesthetic or tactile feedback, despite generally they also involve some data capture capability. In this paper, we prefer the smart gloves name because, although it is the second more used name after “data gloves”, it encompasses the variety of purposes and capabilities in a better way.

Despite the many years devoted to the development of smart gloves, failures to satisfy the complex requirements have been continuous and this device has not become mainstream yet. In any case, the research focused on smart gloves has not decline and during recent years, there has been a growing interest, particularly in the commercial area. The great advances on related technologies, such as wearables and HMDs, have fueled the emergence of new initiatives. Nowadays, there exists a good number of commercial smart gloves and, more interestingly, many pieces of research are being developed based on them. At this point, a main problem is to be able to analyze the features of the different gloves to decide the most appropriate one for a certain application. The goal of this paper is to offer a classification and analysis of existing commercial smart gloves, distinguishing among the different goals and providing a common basis for the decision making.

The rest of the paper is organized as follows. The next section describes the hand anatomy and possible movements that can be produced and captured by the smart gloves. Then, Section 3 introduces the related work, focusing on other surveys and reviews performed about this topic along the years. Next, the method followed to carry out this review based on PRISMA is described. Section 5 introduces the 24 commercial smart gloves identified and Section 6 analyzes them considering 3 main capabilities: hand and finger pose estimation and motion tracking, kinesthetic feedback and tactile feedback. Section 7 reviews the main application areas of these gloves, based on the scientific literature and on the info provided by smart gloves vendors. Finally, Section 8 provides a discussion about existing smart gloves and Section 9 presents the conclusions of the paper.

## 2. The Human Hand

The features of smart gloves are closely related to the anatomy and physiology of the human hand. The concept of degree of freedom (DoF) is particularly important [14,15,16], which refers to the different basic movements that can be performed with the hand and fingers. More complex movements can be performed as combination of basic ones.

Before considering the DoF, it is important to have a good knowledge of the human hand anatomic structure, see Figure 1. A hand is made up by five fingers. Each finger, except the thumb, has three bones (distal, intermediate, and proximal phalanges), and three joints: meta-carpophalangeal (MCP), proximal-interphalangeal (PIP), and distal-interphalangeal (DIP). The thumb has two bones (distal and proximal phalanges) and two joints: meta-carpophalangeal (MCP) and inter-phalangeal (IP). Nevertheless, the thumb has an additional mobile joint: the trapecio-metacarpal (TM).

Regarding movement, the human hand can be modeled with 23 DoF [14]: four in each one of the four fingers, four in the thumb and three in the wrist.

For each finger, except the thumb, PIP and DIP joints can perform an extension/flexion (E/F) movement, while MCP joint can perform E/F and adduction/abduction (A/A) movements, see Figure 2. In practice, depending on the subject, some movements at certain joints cannot be performed independently. For example, many people cannot perform DIP E/F without performing PIP E/F. By the contrary, other movements can be produced by injuries, such as hyperextension and supraduction, but they are not considered.

The thumb has 6 DoF: the IP and MCP joints have an E/F movement and the TM has E/F and A/A movements. Thumb E/F and A/A movements are represented in Figure 3. Active A/A of the thumb MCP joint is limited, considered accessory motions and therefore we do not consider it. Notice these movements are not rectilinear in each of the axes separately, but they are usually carried out jointly, resulting in complex rotational movements.

The wrist provides three more DoF to complete 23 DoF for the whole hand, see Figure 4:E/F or pitch. Extension is the dorsal tilt movement where the hand approaches the back of the wrist. Flexion is the palmar tilt movement where the hand approaches the anterior aspect of the wrist.A/A or yaw. Abduction is the radial or lateral deviation movement where the hand moves away from the midline of the body. Adduction is the ulnar or medial deviation movement where the hand approaches the midline of the body.Pronation/supination (P/S) or roll. Pronation is the internal rotation movement from a neutral position, so that the hand rotates until the back of the hand is facing up (position to catch bread). Supination is the external rotation movement from a neutral position, so that the hand rotates until the palm of the hand is facing upwards (begging position).

The hand can develop other movements, such as the palm bending, but these movements are less important, particularly related to user interaction. Usually, smart gloves are not developed to detect them.

## 3. Related Work

As it has been introduced, smart gloves have a history of more than 40 years. During this time s everal surveys have been published in the scientific literature, most of them during the last years, see Figure 5.

The first survey about glove-based input and electronic gloves was published as early as 1994 [17]. Hands were already considered as the natural way of human interaction with the world, in contrast to the common way of interaction with computers constrained by “*clumsy intermediary devices such as keyboards, mice and joysticks*” [7]. The goal, at this time, was to collect data about the movement and pose of the hand and fingers, naming them as “data gloves”. Some glove devices were already commercialized, mainly related to the needs of the video game industry: The Visual Programming Language (VPL) Data glove (VPL Research, San Francisco, CA, USA) considered as the first data glove appeared in 1987; the Exos Dexterous HandMaster (Dexta Robotics, Hong Kong, China); the Mattel Intellivision Power Glove (Mattel, Inc., El Segundo, CA, USA) as a low-cost version to be used as a control device for the Nintendo video game console in 1989; the CyberGlove from Virtual Technologies (Maumee, OH, USA); and W Industries’ (Houston, TX, USA) Space Glove. This first survey was focused on the hand-tracking features of the gloves, based on three technologies: optical, magnetic and acoustic. Most gloves, for example the VPL Data Glove, were based on the use of optical fibers along the fingers, attenuating the light they transmit when the finger flexion bends the fibers. In other cases, they used Hall-effect sensors as potentiometers at the finger joints that were also able to measure the bending. Despite the existence of these devices as commercial products, this survey concludes that the area of glove-based input was at its infancy. Features such as haptic feedback or wearability were not considered at all.

The next survey in the literature about smart gloves was published fourteen years later, in 2008 [11], reflecting a slow progress in the technology. This survey used the name “*Glove-based systems*”, described as “*composed of an array of sensors, electronics for data acquisition/processing and power supply, and a support for the sensors that can be worn on the user’s hand*”. Typical gloves at that time were described as “*a cloth glove made of Lycra where sensors are sewn*”. They had limitations in the form of portability, as they required wired physical connections, limited haptic sensing and naturalness of movement. At this time, actuators and kinesthetic feedback are considered as glove accessories and not as an essential feature. Thirty different gloves are described in this survey, both commercially available and prototypes, ranging from 1978 to 2008 and classified in three stages in the evolution:
Early research. Gloves equipped a limited number of sensors, hard wired and developed to serve specific applications, never commercialized.Data glove-like systems. These shared three basic design concepts: (i) they measured finger joint bending using bend sensors; (ii) they used a cloth for supporting sensors; and (iii) they were usually meant to be general-purpose devices. Several commercialized products are referenced: VPL Data Glove in 1987 by VPL Research, Inc.; the Power Glove by Mattel Intellivision; Super Glove in 1995 by Nissho Electronics (Tokio, Japan); the P5 Glove in 2002 by Essential Reality, LLC (Shelby, OH, USA). Other devices mentioned were the Space Glove, CyberGlove (CyberGlove Systems LLC, San José, CA, USA), Humanglove (Humanware, Pisa, Italy), 5DT Data Glove (Fith Dimension Technologies, Orlando, FL, USA), TCAS Glove (T.C.A.S. Effects Ltd., city, state abbrev if USA, country), StrinGlove (Teiken Limited, Osaka, Japan) and Didjiglove (Didjiglove Pty Ltd., Melbourne, Australia). Interestingly, CyberGlove has been a major reference in the domain but was recently discontinued (see Section 4) and 5DT is still active (see Section 5).Beyond Data Gloves. This category gathers devices with no cloth, such as rings, and using new sensor technologies (e.g., infrared LEDs and changes in skin coloration, accelerometers, LED scanner), trying to support specific applications, particularly alphanumeric character entry.

The next survey, published nine years later in 2017 [18], was focused on wearable haptics for the finger and the hand and not on the data capture capabilities. In contrast to grounded and bulky haptic devices, the paper highlights the efforts to provide “*wearable haptic systems for the fingertip and the hand, focusing on those systems directly addressing wearability challenges*”, such as the CyberGrasp exoskeleton (CyberGlove Systems LLC, San José, CA, USA) or the Rutgers Master (Burdea, Romania), but yet too complex and expensive in consumer terms. This survey provides a classification distinguishing among the type of tactile stimuli provided to the wearer (kinesthetic, pressure, contact, vibration, curvature, softness); area of the end-effector (fingertip or whole hand); technologies (e.g., DC motor, air jet nozzles, servo motors, voice coils, vibrating motors, pneumatic actuators, dielectric elastomer actuators) and level of wearability (weight and dimensions). It analyzes 23 fingertip devices and 23 haptic devices for the whole hand. These devices were prototypes described in papers available in the scientific literature and were used to provide kinesthetic and tactile (pressure, contact or vibration) stimuli. No commercial devices are analyzed.

The next year, 2018, haptic gloves were reviewed in [19], including a good number of commercial products. This survey distinguishes among traditional gloves, thimbles and exoskeletons. In total, 13 different devices were analyzed, considering that although belonging to different classes they all share the same objectives and constraints. In more detail:The “*traditional glove*” refers to “*a garment made of some sort of flexible fabric, which fits the shape of the hand and lets the fingers move individually”. “The sensors to measure the flexion of the fingers and the actuators to apply a feedback on the skin or skeleton are either sewn within the fabric or fixed on the outside of these gloves*”. As examples of this type, paper authors include the Avatar VR by Neurodigital Technologies (Seville, Spain), which evolved to become the Sensorial XR, included in this paper. Other examples described have been discontinued, such as Maestro (Markham, ON, Canada).A “*thimble*” is “*a configuration with an actuator attached to a fingertip. It is possible to combine several thimbles in order to provide feedback on several fingers at the same time. In such a way, a function similar to that of a haptic glove can emerge*.” In this paper, we do not consider these devices or their combinations as commercial gloves. In any case, some commercial solutions, such as Polhemus (Polhemus, Colchester, VT, USA), could be considered in this category.Exoskeletons. “*An exoskeleton is an articulated structure which the user wears over his/her hand and which transmits forces to the fingers*…” enabling in this way the provision of kinesthetic feedback. Examples of commercial exoskeletons are: CyberGrasp, HaptX (HaptX Inc., San Luis Obispo, CA, USA), Dexmo by Dexta Robotics (Hong Kong, China), VRGluv (VRgluv, Georgia, GA, USA), Sense Glove DK1 (Sense Glove, Delft, The Netherlands) and HGlove (Haption SA, Soulgé-sur-Ouette, France). Some of them have been discontinued recently.

Next year, in 2019, three papers were published that can be considered as smart gloves surveys, two of them by the same authors: Wang et al. The first one was focused on force feedback gloves [20]. This includes a detailed classification featuring motion tracking and kinesthetic feedback capabilities. Specifications used to quantify the performance of motion tracking are: degrees of freedom (DoF), motion range, sensing accuracy and update rate. For kinesthetic feedback, the following specifications were used: dimension (actuated DoF), range of applicable force, resolution and dynamic response of feedback forces. This work analyzed several research prototypes and two commercial gloves: CyberGrasp (CyberGlove Systems LLC, San José, CA, USA) and Dexmo (Dexta Robotics, Hong Kong, China). Gloves were also classified according to the location of the actuation into four sub-categories:Ground-based systems. The base is fixed on the ground or a desk. From our point of view, these are not real smart gloves.Dorsal-based systems. It is a wearable exoskeleton system grounded to the back of the hand.Palm-based systems. Grounded to the users’ palm. The force is provided directly between the fingers and the palm to simulate palm opposition type grasping.Digit-based systems. Grounded to the digit, provides forces directly between the finger and the thumb to simulate pad opposition or precision type grips.

The second review paper by Wang et al. focused on haptic displays for VR [21]. This distinguishes among desktop haptics, surface haptics, and wearable haptics. Haptic gloves are considered in the case of wearable haptics, providing both force and tactile feedback to fingertips and the palm. This work ref. [11] commercial gloves providing motion track, force feedback and tactile feedback. Nevertheless, as this work is about haptic displays in general, it does not provide a detailed analysis of the features of the commercial gloves.

Also, in 2019, a survey about wearable technologies for hand joints monitoring for rehabilitation was published [22]. This survey introduces several smart gloves, some of them commercial, analyzing their capability to support human hand rehabilitation. The different gloves are classified in accordance to their technology, distinguishing among the following ones:Flex sensor-based technologies, referencing the commercial ones: CyberGlove III (CyberGlove Systems LLC, San José, CA, USA), 5DT Data Glove (Fith Dimension Technologies, Orlando, Fla., USA), X-IST Data Glove (SouVR International Trading Co. Ltd., Beijing, China) and DG5 VHand 2.0 Data Glove (DGTech Engineering Solutions, Bazzano, Italy).Accelerometer based technologies, referencing the commercial ones: KeyGlove (Jeff Rowberg, Roanoke, VA, USA) and AcceleGlove (Whashington, DC, USA).Hall-effect sensor-based technologies, referencing the commercial one Humanglove.Stretch sensor-based technologies. No commercial gloves are referenced.Magnetic sensor-based technologies. No commercial gloves are referenced.Vision-based technologies. These cannot be considered as smart gloves, because they are based on the use of external cameras and gloves painted with different colors to facilitate the recognition of the fingers and hand position.

Last year (2020), a survey about hand pose estimation with wearable sensors and computer-vision-based methods was published [23]. It analyses various types of gloves and computer-vision-based methods proposed for hand pose estimation in recent years. This paper introduces a sensor taxonomy for gloves distinguishing among bend (flex) sensors, stretch (strain) sensors and other types, such as inertial measurement units (IMUs) and magnetic sensors. Besides, references to research and commercial gloves and the type of sensors used are also included. For the commercial ones, they mention: CyberGlove III, 5DT and Hi5. In addition, this survey also describes computer-vision-based methods, based on the use of cameras to capture RGB images or commodity depth sensors, which enable them to create depth maps. A major problem for these methods is occlusion as they rely on line-of-sight observation, namely, they are very likely to be blocked or partially blocked while doing activities and manipulating objects. As a conclusion, this is a main argument in favor of developing smart gloves. This survey does not consider any kind of haptic feedback features in gloves.

Also in 2020, a paper about tactile feedback in VR was published [24], where four commercial ones are referenced: Manus VR (Manus Machinae B.V., Geldrop, The Netherlands), VR Free (Sensory Ag, Zürich, Switzerland), Plexus VR (Digital Kinematics, London, UK) and Dexmo VR (Dexta Robotics, Hong Kong, China). In any case, this cannot be considered as a survey as the gloves are neither described nor analyzed.

No one of the existing surveys has been done following a review methodology, such as PRISMA. In addition, previous surveys are focused on some specific kind of capability, such as hand or finger tracking, but not on the variety of capabilities. In other way, some of the surveys do not involve gloves only, but also other kind of devices, such as thimbles or rings. More importantly, no survey has focused on commercial devices exclusively. Therefore, taking into account the current interest on these devices, we consider a survey about smart gloves an interesting topic.

## 4. Methods

The present survey of smart gloves is aligned with the PRISMA guidelines for systematic reviews and meta-analysis [25]. As this is the first systematic review on this topic, the review protocol has not been registered. Data was sourced from published articles, see Figure 6. The primary databases searched were MDPI, Elsevier, Springer, Taylor and Francis, and IEEE Xplore. As a secondary source, Google Scholar was used. The primary keywords were “cyber glove”, “data glove”, “force-feedback glove”, “glove-based system”, “haptic glove”, “sensory glove”, “smart glove”, “virtual glove” and “VR glove”. In cases where the plural form provided additional relevant resources, it was also used. The inclusion and exclusion criteria were characterized by a title and abstract screening followed by a full-text and abstract screening process. Despite a large number of references were found in the primary and secondary sources, the screening process was quite straightforward because we focused on commercial devices. Many papers were discarded because no mention of commercial smart gloves was found. Next, 598 were discarded because they just mentioned some commercial glove, but not their use or an analysis of them.

Searches were limited to the period 2015–2021 and only papers in English language were included. The inclusion criteria were to be an active commercial wearable device for the full hand. Therefore, the exclusion causes were as follows:Non-commercial devices. Many papers in the literature describe the development or proposal of smart gloves, based on the use of special sensors, actuators or developed towards a specific purpose. Some devices commercialized in restricted contexts were neither included, such as the NuGlove (Anthro Tronix, Silver Spring, MD, USA), only available for military purposes.Recently discontinued devices. There are many commercial smart gloves that are no longer available in the market. The most relevant case is related to the four different solutions (CyberGlove, CyberTouch, CyberGrasp and CyberForce) produced by Cyberglove Systems, discontinued in 2019. Remarkably, this company has a long history in the smart gloves domain, including 446 references in the literature since 2015. Other examples of recently discontinued products are: HaptX, Keyglove, HumanGlove, Plexus, Maestro Gesture Glove, Teslasuit Gloves (VR Electronics Ltd., London, UK) and VRGluv. In many cases, the discontinued gloves evolved to new products as different companies, such as the AcceleGlove that evolved to become NuGlove and GoGlove. It is more common that a company issues new version of its device under a different name, such as Neurodigital with Gloveone and Avatar VR, previous to Sensorial XR. In the case of Peregrine by IronWill, despite not currently commercialized yet, it is announced that they will be available in the second semester of 2021.Non-wearable devices. Some devices are not really wearable, but they are connected by strings or rigid structures to some ground system or device to be carried on by the user. For example: ExoHand by Festo (Esslingen, Germany) is attached to a pneumatic system, HGlov by Haption is also connected to a rigid structure, Glohera Sinfonia (Lumezzane, Italy) and Esoglove (Roceso Technologies, Bukit Merah, Singapore).Devices that do not allow one to use the hands freely. For example: Microsoft Haptic Pivot and Valve Index controllers. Commercially these devices are very relevant, as they are linked to some of the main companies in the XR domain. Nevertheless, they are based on the use of vision-based solutions to recognize the hand and finger pose and movement. This limits user movements as the user can clash with objects while moving the hands.Not full-hand devices. There are some proposals that consider just some fingers or a part of the hand, such as rings or thimbles, or only the wrist. For example: Fingertracking by ART (Advanced Realtime Tracking, Bayern, Germany), Polhemus, and EXOS wrist DK2 by Exiii (Exiii Inc., Tokyo, Japan).Devices that do not provide real smart gloves capabilities. For example, some gloves just detect touches at specific hand/finger locations, such as the touch of the fingertips of the thumb and index finger. They are intended to be used as a kind of remote control, such as Saebo Glove (Saebo, Charlotte, NC, USA) and GoGlove (GoGlove, Los Angeles, CA, USA); or for rehabilitation purposes, such as the MusicGlove (Flint Rehab, Irvine, CA, USA). In other cases, the purpose is to quantify the pressure applied to and exerted by the hand, such as the TactileGlove (PPS, Boston, MA, USA). In any case, these gloves have some capability to enable hand interaction in XR environments.

In total, 29 devices have been discarded. They are referenced in Table 1.

## 5. Results

This section introduces the commercial smart gloves identified in the survey performed in accordance to the method described in the previous section. The obtained gloves are shown in next Table 2 and Figure 7. Despite the numerous discards, the number of gloves is quite large: 24. The table includes information about the country of the company, URL, glove type as described in next Section 5.1, capabilities supported and price. Three main capabilities are recognized (see Section 5.2): hand and finger pose estimation and motion tracking, kinesthetic or force feedback and tactile feedback. Additionally, Section 5.3 analyses issues related to ergonomics and wearability.

Smart gloves companies can be found all around the world, mainly in USA and Europe, but also in China, Russia, Israel and New Zealand. The discarded gloves include products from other countries, such as Japan or Canada, showing the global interest in this type of device.

Many companies have different versions of their gloves active: 5DT, Dexmo, Synertial, Manus, Nansense, Noitom, SenseGlove and VMG. In many cases, they provide the same glove with a different number of sensors to enable the tracking of more DoF, such as 5DT, Cobra Glove, Dexmo, Nansense and VMG. In other cases, companies offer gloves with different capabilities, such as Manus and VMG that offer gloves supporting just tracking and positioning and another model offers kinesthetic feedback. As particular cases, the Chinese Noitom company sells the Hi5 VR and the Perception Neuron Studio Gloves (despite the fact different URLs may be shown in Table 2) and SenseGlove from the Netherlands have two gloves with similar features based on different technologies. Finally, some of the gloves are not the first ones developed by the company, but evolutions from previous models, such as Manus and Neurodigital devices.

Many of the companies producing smart gloves are startups that have the gloves as their only product. Some examples are: CaptoGlove, ManusPrime, SenseGlove or SensorialXR or VRGluv. Notice, VRGluv was launched in 2017 at a crowdfunding web page, but it has since been discontinued. In other cases, smart gloves companies are involved in the motion capture business, and they have other products such as body suits to recognize the human body movements. This is the case of Cobra Glove, Nansense, Perception Neuron and Rokoko. In many of these cases, the gloves are not sold separately. The New Zealand stretchsense is based on a special stretch sensor technology that is also included in other products of the company.

Regarding prices, it is important to notice that these are approximations. We have tried to indicate the cost of the pair of gloves. Nevertheless, there are some variations depending on possible accessories included, such as batteries or connectivity options, and also related to special guaranties or licenses. In any case, the prices are generally above €1000.

### 5.1. Glove Types

Commercial smart gloves are usually classified in two main categories: exoskeleton and fabric. In addition, when we pay attention to Figure 7, it is possible to identify other distinguishing features, such as strips and open fingertips. As a result, we propose the following classification:Exoskeleton. This refers to a structure located in the back of the hand involving some strings or rigid links attached to the fingers. They are used to provide kinesthetic feedback to the hands.Fabric. This refers to a piece of fabric that covers the full hand and fingers. Inside the fabric, some sensors and actuators are included to perform the desired capability.Strips of fabric, plastic or other materials. Some smart gloves do not cover completely the skin of the fingers and hand. Instead, they have just fabric, plastic or other materials in the locations where the sensors and actuators are located. This kind of gloves can facilitate the fitting to different hands and fingers shapes and forms.Open fingertips. Some smart gloves have open fingertips. This feature facilitates the use of touch screens and other activities where the finger sensitivity is important. It can also enable a better glove fitting.

### 5.2. Capabilities

Smart gloves can be used for different purposes and the following ones are generally recognized [20,26,27]: Hand and finger pose estimation and motion tracking. This capability is also known as “*hand posture reconstruction*”, “*hand movement tracking*” and “*hand movement synthesis*”. This involves the capability to measure the position and movements of the fingers and the whole hand, as described in Section 2. Motion tracking is necessary to detect user’s manipulation gestures and to drive the motion of a hand avatar in virtual environments. Another issue related to hand and finger pose estimation is gesture detection. Gestures can be determined from hand and finger tracking, but there are some gloves that also determine gestures by other means. For example, gestures such as the joining of two fingers can be detected with sensors located at the fingertips. High DoF and large motion range are required for recognizing dexterous manipulation and grasping. Furthermore, high resolution and update rate are required for simulating fine manipulation and actions such as the pushing of a button. Specifications used to quantify the performance of motion tracking are [20]: DoF, motion range, resolution and sampling/update rate. In addition to position-motion, it would be also very interesting to measure the force exerted, but this is something more complex that is not supported by existing commercial gloves.Haptic feedback. This is related to the human perceptual system which includes various kinesthetic and cutaneous receptors in our body, located in the skin, muscles or tendons. Haptics technology simulates the sense of touch in computing [28] and involves two different features [19]:○Kinesthetic or force feedback. Referred to provide the impression of movement and resistance through the muscles, like the feeling of weight, inertia, or resistance. It involves the reproduction of movements and resistances by means of actuators, such as electric motors, to exert specific forces in the hand and fingers. This can be used to simulate the touch of immovable objects such as walls, the grasping of virtual objects, the use of triggers, etc. To this end, a sufficient range of force/torque and magnitude is required. In addition, it is also important to have a good force resolution in order to simulate subtle changes and contact with small objects. The following features are important for the kinesthetic feedback [20]: dimension (actuated DoF), range of applicable force (e.g., maximum fingertip force), resolution and update rate. Notice that kinesthetic feedback requires hand tracking, but not vice versa.○Tactile feedback. Tactile feedback devices provide input to the user skin [29] to recreate different sensations, such as shape, texture, thermal, smoothness, etc. In haptic devices, this is achieved through different elements [21], such as mechanical vibration, surface shape changing and friction modulation. In the case of commercial smart gloves, mechanical vibration is the option, involving the use of motors, linear resonant actuators, voice coils, solenoid and piezoelectric actuators.

### 5.3. Ergonomics and Wearability

Ergonomics and wearability concepts are related to the usability of the smart glove. It is desirable that smart gloves are comfortable to wear, easy to put on and off and do not limit or restrict the activities performed by the user. In addition, to avoid users’ fatigue, gloves should be as lightweight as possible, including its battery and controller. Another requirement is related to safety, as gloves should never injure the user even in the occurrence of system failures, particularly in case of kinesthetic gloves.

There are many features that can be considered related to the ergonomics and wearability: size, weight, power consumption, etc. Kinesthetic gloves [20] usually involve the method of mounting the gloves to the human hand, the way of transmitting forces and torques to fingers, that makes them especially complex. From all the possible features, taking into account the information available from the different products, we have gathered the following ones, see Table 3:Size. Gloves should fit an arbitrary size and form of the hand or should be easily adaptable. This constraint can be addressed by offering a selection of sizes within a certain working range. In any case, some gloves just provide a unique size.Weight. Gloves should not be heavy. From the data collected, weight varies from 50 to 300 g.Battery and Autonomy. Most gloves include some kind battery. Autonomy varies between 2 and 10 h, but this depends on the operation level.Wireless. The ability to work without cables improves freedom of movement, especially in cases where manual activity measurement is required. Most gloves include some kind of wireless technology, whether Bluetooth or Wi-Fi based. Most models also include a cable connection and, in some cases, (e.g., Exo Glove, MoCap Pro SuperSplay) an SD Card to store the data captured.

## 6. Analysis

This section analyzes the identified commercial gloves paying attention to the three main capabilities recognized: hand and finger pose estimation and motion tracking, kinesthetic feedback and tactile feedback. Therefore, next subsections describe features and capabilities of the smart gloves under each one of these capabilities.

### 6.1. Gloves for Hand and Finger Pose Estimation and Motion Tracking

The main smart gloves capability is hand and finger pose estimation and motion tracking. This can be achieved using different kinds of sensors (e.g., IMU, stretch and strain sensors) located on gloves or using visual-based solutions, such as leap motion based on infrared sensor [29]. Visual-based solutions present important problems, such as occlusion, that cannot be easily solved, as they cannot capture hand and finger out-of-sight movements. In some cases, mixed solutions are proposed, with gloves including especial markers that can be easily recognized by image sensors (e.g., the Vincon motion system described at vicon.com last accessed on 27 February 2021). These systems offer higher precision and faster measurements than the markerless vision-based ones.

The approach to capture movements through sensors generally involves a mapping that goes from the sensor output to hand and fingers joint angles. Some devices will allow direct measurement of all finger DoF. For others, inverse kinematics may be used, for example, to calculate internal joint angles from the position of fingertips relative to the palm. Relationships between the DIP and PIP angles can also be enforced to reduce the active number of DoF [30].

Table 4 includes the commercial smart gloves that support this capability, indicating technology and sensors included. All the gloves identified as smart gloves support this capability, as long as it is basic to support the other ones. Most gloves employ sensors from three categories: IMU, bend (flex) sensors and strain (stretch) sensors. Other proposals in research have used magnetic sensing [31], capacitive sensors [32], or electromyography (EMG) [33] for gesture recognition. For a complete overview, we refer to the existing surveys [11,22]. In more detail, and regarding commercial smart gloves these are the technologies involved:IMU. This device can measure acceleration, rotational speed and orientation [27]. It is made up by several sensors: a 3-axis accelerometer, a 3-axis gyroscope and a 3-axis magnetometer. These sensors are relatively low cost and can have very high sampling rates. A main issue with IMU is related to drift. The data collected from sensors is integrated from a known starting configuration. As a result, errors are accumulated, and even small variations can lead to large errors over time. Another major drawback of IMU in the smart gloves’ context is their rigidity and bulkiness compared to the size of human fingers. In any case, the use of IMU is very common to estimate hand position and movement, mainly when the goal is related to motion capture, such as Cobra Glove, Exo Glove, Hi5, Nansense R2, Perception Neuron, Sensorial XR, Senso Glove DK3 and VR Free. A particular case is Rokoko, which uses a IMU without a magnetometer to ensure immunity from magnetic distortion. Indeed, usually smart gloves vendors based on other technologies highlight that their gloves are immune to magnetic fields, in contrast to IMU-based gloves. Finally, in many cases, external IMU can be attached to the gloves over the wrist to estimate the hand position and movement in the space, as it is shown at column “External IMU” in Table 4. Usually, commercial trackers provided by VR vendors, such as Oculus or HTC Vive, are attached. These devices help to make the gloves more modular, but they need to be calibrated in conjunction.Bend (flex) sensors. These are piezo resistive elements that change their resistance as they are bent or flexed, creating variations in the transmitted electrical signal. Such variations can be measured and mapped to changes in joint angles. The ideal solution involves the availability of linear behaviors, where the change in voltage can be linearly related to the finger bend, as this would facilitate the mapping between the signal and the joint angle. Bend sensors should be placed in the gloves in such a way that they are exactly in the location of the joints of interest, such as the interphalangeal joints. In such a position, a one-to-one mapping between the joint blend and the sensor reading could provide an accurate measurement. It is particularly difficult to accurately measure joints involved in abduction and adduction movements [27]. Another problem for bend sensors is their short lifespan, as the continuous bending makes them to break quite fast. In any case, bend sensors have been extensibility applied in commercial smart gloves like the 5DT, Capto Glove, Forte Data Glove, HandTutor, Manus Prime II, Rapael, VMG and VR Free.Strain (stretch) sensors. Strain sensors provide a changing signal as they stretch. This effect can be obtained from resistance or capacitive elements. Capacitive sensors can be relatively smaller, facilitating the integration of a larger number in a reduced area, such as in the case of The StretchSense MoCap Pro.Rotational sensors. There exist some gloves that include mechanical rotational sensors, such the Dexmo exoskeleton, or rotational encoders, such as the SenseGlove DK1. These sensors are able to convert the angular position of a shaft to a signal. They are quite bulk, but they can be embedded in these gloves because their exoskeleton structure facilitates it.Hybrid approaches. There are several approaches that combine different technologies. The idea is to solve the issues present in some approaches and to take advantage of the strengths. For example, to mitigate the drift of IMU, the poor abduction tracking of bend sensors, or the interference of magnetic fields. Rokoko combines IMU with magnetic tracking. Manus Prime II combines bend sensors with IMU. SenseGlove Nova and VRFree combine IMU with a visual-based method.

Some commercial gloves have not been included in Table 4 because we have not been able to get the information about their sensor technology. This is the case of Anika Rehap and Cynteract. Both of them are focused on supporting hand motor skills coordination and rehabilitation, and therefore the capture of specific DoF is not a main concern.

Taking the information provided by the vendors, number of sensors and technology, we have compiled Table 5 showing the DoF per hand for the commercial smart gloves. This table includes not just the global number of DoF, but also the precise DoF for the fingers and the hand according to the reference model described in Section 2. This table includes many annotations regarding the different ways in which DoF have been estimated, because information provided by vendors is not completely clear in all cases.

A main appreciation needs to be noted regarding IMU-based smart gloves. In this case, the relationship between sensor location and DoF involved is not straightforward. This kind of sensor measures the global movement. Therefore, a reference to a fixed point needs to be established to estimate the actual DoF performed. In case of bend or stretch sensors, the relationship between the movement and the DoF is clearer.

More info about the tracking and positioning capabilities of the smart gloves would be interesting, such as motion range, resolution and sampling/update rate. Nevertheless, many vendors do not publish this data, or such data is not provided in a standardized way. For example, in some cases, raw sensor values are provided, while in other cases a normalized value in the range 0-1 is facilitated. Similarly, some DoF are usually not provided as a result of a sensor measure, but as an interpolation of related measures.

### 6.2. Gloves for Kinesthetic Feedback

Actuator technologies for kinesthetic feedback can be classified into two modes [20]:Passive actuation principle or impedance control. This involves the application of a resistance to the hand and fingers in accordance with their movement. An impedance glove has to detect the movement of the fingers (sensing of the motion) and to apply a resistance force to provide kinesthetic feedback. Therefore, these gloves only provide feedback when the user tries to move, but not when the user’s hand remains motionless. This technology is intrinsically safe as there is no chance of harm for the user, even in case of system failure. Some technologies used to provide passive kinesthetic feedback are: magnetorheological fluids (MRFs), brakes, clutches and springs, and pneumatic jamming.Active actuation principle or admittance control. In this case, the smart gloves apply force to fingers to make them move. This technology can provide not only active motion, but also resistance force or torque. The advantage of the active solution is to provide active control and simulate active force/motion output in a high update rate, while its disadvantage is potential risk of injuring the fingers in the event of a system failure. To avoid this possible failure, most of the active gloves limit the maximum output force to about 10 Newtons. Some technologies used are: DC servo motors, hydro pump or valves, pneumatic pump or valves, dielectric elastomer, etc.

In the literature, most force-feedback gloves are using the passive control principle [20]. Similarly, commercial smart gloves providing kinesthetic feedback also develop the passive mode, see Table 6. Notice, just three among all the selected gloves provide this feature. Two models have been recently discontinued: VR Gluv and TeslaSuit Gloves. In this table, it is shown the different technologies used and the actuated DoF. Despite the differences in technologies, it can be observed a similar performance in terms of actuated DoF and force exerted. The discontinued VR Gluv provided 10 actuated DoF, with two actuated points per finger, in the distal and proximal phalanges.

### 6.3. Gloves for Tactile Feedback

A good number of smart gloves have focused on tactile feelings at hands and fingers, see Table 7. The scientific literature describes many attempts to provide tactile feedback involving different features [18], such as perception hardness (hard/soft), warmness (warm/cold) macro roughness (uneven/flat), fine roughness (rough/smooth), and friction (moist/dry, sticky/slippery). Several technologies have also been explored, mainly electric motors, but also microfluidic arrays or electrostatic attraction. In case of the commercial solutions, vibration is the unique technology in use. Particularly, it is rather common to include linear resonant actuators (LRAs), similar to the vibrating motors used in game controllers and smartphones. In most cases, the sensations produced by the tactile actuators are not specified, but they can be programmed in accordance with the desired application.

Table 7 gathers the nine out of 24 commercial smart gloves providing tactile feedback. As it can be observed, they share many features. Most of them involve the use of LRA over the fingertips, thumb and palm. Usually, they are programmable, and, in some cases, they involve the provision of specific feedback, such as the detection of collisions, textures, button clicks, etc. The VR Free Haptics gloves have not been included because no information about sensors and features was found.

### 6.4. Other Features

In addition to the main capabilities described in the previous sections, commercial smart gloves include some capabilities to facilitate the user interaction:Pressure sensors. Several gloves include pressure sensors to measure the force exerted. The CaptoGlove includes this kind of sensor on the thumb’s fingertip able to perceive pressure from 100 g. to 10 Kg, oriented to detect specific gestures, such as the pushing of a button. The Senso Glove also integrates pressure sensors to measure the grip pressure. VMG 13, 30, 35 and PS gloves also include pressure sensors, one per finger, which can be used to emulate a mouse/keyboard or develop custom actions. The Cynteract also has a pressure sensor.Screen interaction. Some gloves include a special fabric at the fingertips to facilitate the interaction with touch screens, such as smart phones and table ones. The CaptoGlove also has this feature for the fingertips of the index finger and thumb.Touch points. Some gloves include conductive points at certain points of the fingers or hand that are activated when the user touches them. The sensorial XR has conductive zones that enable users to trigger specific customized actions. The Peregrine, a previous version of the future Peregrine VR glove, includes 17 touch points: five points per finger (three on the pinky), each makes a keystroke when touched by the thumb tip contact. The MusicGlove includes six Nora-LX Conductive Metallized Fabric (Ni/Cu plated plain weave fabric) located on all five fingertips and one on the proximal interphalangeal joint on the lateral aspect of the index finger. When the lead of the thumb touches any of the other five leads, an electrical connection is closed which is then registered by the computer as an event. Notice we discarded some devices as smart gloves because they only provide this feature (see Section 3).

## 7. Application Areas

The applications and projects in which the commercial smart gloves have been used as well as the motivation for their use give us an idea of the capacity for measurement or response that can be expected from them. For this reason, this section was born, not only to have a starting point for possible uses and applications but also to learn more about gloves in their study environments.

During our search for commercial smart gloves, we have come across a multitude of fields of application where they are used, see Table 8. Many gloves provide a new range of applications in gaming, industry, surgery training, rehabilitation and education. Throughout this section, we will try to classify these fields of application, bearing in mind that in many cases, these fields are not independent, but rather overlap or even more generic ones include more specific ones. For example, some gloves with haptic feedback are being proposed for hand and finger rehabilitation or surgery training, both fields belonging to a medicine scope. Another example would be the communication by means of gestures (motion capture) with a person with some disability during an emergency (medicine and health care). Taking this into account, we have classified the fields of application into six categories, which we will detail in the following subsections, together with those gloves most commonly used in each of them.

### 7.1. Medicine and Remote Health Care

This category encompasses all those gloves that are used in the field of medicine in general and in particular in issues directly related to remote health care. In this category, we find two main areas of application, such as remote manipulation and rehabilitation:In the case of remote manipulation of robot arms/hands, its main application is related to teach the fundamental skills of robotic surgery to novice and experienced surgeons (usually through simulations) and that of performing surgery on real patients.In the field of rehabilitation, we can find two main types of applications. Firstly, those aimed at making a diagnosis of the functionality of the hand, that is, to check if the movements of the hand when carrying out certain actions are adequate (for example, for patients who have suffered strokes). Secondly, we have applications dedicated to actively treating mobility problems, making users perform a series of exercises (usually through the development of specific video games for it). In many cases, they are focused on the recovery of hand mobility after a stroke. In this field, one of the most outstanding smart gloves is Rapael. Some smart gloves have been specifically developed for rehabilitation purposes: Anika Rehap, CaptoGlove, Rapael, Handtutor, MusicGlove.

### 7.2. Motion Capture

Gloves whose main purpose is to capture movements belong to this category. Obviously, capturing the movement of the wearer’s hand is a requirement of virtually any application that uses the type of gloves discussed in the paper. However, in this category we will refer above all to two main areas of application: (i) motion capture that is carried out mainly for use in animations, digital avatars, etc.; and (ii) the specific analysis of hand gestures.

Motion capture has the field of entertainment as one of its main applications. This type of capture has been used profusely in recent years in the world of film and television, music concerts, as well as in video games, largely for the recreation of virtual characters. Said movement of the hands using gloves is usually accompanied in certain cases by other elements, such as full-body suits. Some smart gloves have been specifically developed for this purpose, and indeed, they are sold as a part of a smart suit for whole body tracking: Cobra Glove, Nansense, Perception Neuron and Rokoko.

Gestural analysis of hand movements is largely used for communication and sign language, which rely on hand poses that can be relatively more complex and involve close interaction of the fingers. These types of data captures can be applied in multiple areas, such as patient monitoring, virtual and augmented reality navigation and manipulation, home automation, robotics, vehicle interfaces, PC interfaces, and lexicon translation of sign languages, among others [95]. In particular, the use of smart gloves in the field of sign language has acquired special relevance in recent years [96], being 5DT one of the more relevant smart gloves in this field.

### 7.3. Video Games

In this case, we will address exclusively those video games focused on the world of leisure, entertainment, or even sport. We will not take into account other types of video games in the style of serious games, such as those used to support rehabilitation tasks, simulation training, teaching, etc.

The field of video games is eminently transversal, as we see in the rest of the categories, we could include it in practically all of them (video games of one type or another are used in medicine, motion capture, simulators, manipulation of 3D objects, etc.). However, beyond that generic approach, touching on so many fields of application, we wanted to highlight in this category its specific use as an independent field focused on the purely playful side of its application.

### 7.4. Simulation and Training

This category includes all those tasks that are oriented to carry out some type of training or simulation but excluding those related to medicine (since they are dealt with in the previous Section 7.1), and that have a learning component. In another case, they would be considered within the previous video game category.

In most cases, these tasks are related to training and learning in the use or simulation of different types of devices. We also consider here several industries, such as trucking, construction, mining, agriculture, aviation and many more.

In this category we also find gloves used in simulators for music learning, such as Captoglove.

### 7.5. Manipulation of 3D Objects (Both Real and Virtual)

Object manipulation relies on contact between the hand and the object. This contact can involve any or all of the fingers, ranging from the fingertip to the entire length of the finger (even involving the palm). For example, grasping is an important class of these manipulations.

Gloves that are used in the manipulation of 3D objects that are in a totally virtual environment, as well as those that we can find in the real world, will be considered in this category. In the latter case, most of the applications in which gloves are used are related to the remote manipulation of a robot arm/hand (and as was the case with the previous section, we will not take into account in this case those uses related to medicine). We will also take into account in this category the gloves used both for design and for product testing.

### 7.6. XR Applications

This is a general category in which to accommodate those applications that do not have a clearly defined category, or whose entity is not sufficient to constitute an independent category. Therefore, here we can find applications that, from a certain point of view, we could have assigned to any of the previous categories, but that we have preferred not to categorize in such a specific way, or that could directly be assigned with the same weight to more than one category.

Example of this type of application are: transmit sensations of the virtual world with realism (such as textures or even raindrops on the hands); natural and accurate testing of pressures applied to and exerted by the hand; providing spatial guidance in 3D space; manipulation of multimodal data; virtual visits such as a zoo, interactivity in virtual worlds, etc.

## 8. Discussion

For a long time there have been a great interest in the development of smart gloves. They are considered as a natural way for human-computer interaction, particularly in XR environments where the user immersion and embodiment are given a great importance. Such importance has attracted the interest of many companies that have delivered numerous devices along the last years. Similarly, many researchers have attempted to take advantage of commercial devices to solve problems in multiple domains. Nevertheless, on the contrary, current technology seems not mature enough to provide satisfactory results in all the fields considered. There are some smart gloves whose development is mature enough to use in the scopes of medicine, simulation, or motion capture. However, there is still a long way to go in other fields of application such as videogames or manipulation of 3D objects, or even in XR in general. Also, and related to the fields of application, it is difficult to carry out a clear categorization. Except for some specific cases, most gloves, even being designed for a specific application, can actually reach to be used for general-purpose, and as a result the field of application becomes heterogeneous and difficult to define.

Related to the possible applications we consider prices are still high for the general consumer market and this is rather volatile, with new companies launched and other ones discontinued. This is something that can be specially observed during the last years, maybe fueled by the successes of related products, such as smartwatches and HMDs, in the context of startup initiatives. On the one hand, some of new companies described in this paper are still very active, such as Manus VR, SenseGlove, Neurodigital or Sensoryx, while other ones already discontinued, such as Plexus, Teslasuit or VRGluv. On the other hand, companies with a larger lifespan and a significant presence in the market and scientific literature have been recently discontinued, remarkably Cyberglove Systems. In any case, from the number of products identified it is clear the global and increased interest about this type of device, particularly from the view of potential applications.

The current smart gloves market can also be featured by the variety of options and features that make comparison difficult among available products. Firstly, a generally accepted name for the variety of devices does not exist. We opted by the smart gloves, but other names are already used in the domain. Similarly, a variety of glove types exist, being the more recognizable ones the exoskeletons. Secondly, there is not a clear set of features to be supported. Clearly, hand and finger pose estimation and motion tracking should be provided, while kinesthetic and tactile feedback can be optional. Furthermore, other features such as pressure or touch capabilities should be avoided as main capabilities, despite they can be very useful for certain applications. In any case, it would be important to clarify these capabilities in the description of the products. Thirdly, related to the main pose estimation and motion tracking capability, there does not exist any reference model that enables to compare the capabilities of the different products. As a clear example, some companies indicate that their gloves recognize more DoF that the available ones in the hand, a least according to the model described in Section 2. More commonly, it is not clear how many DoF are really supported because the vendors do not provide all the needed information. Actually, different technologies do not provide the same level of performance. IMU based solutions can be used to estimate the pose and tracking, but not to provide a direct measurement of specific DoF such as bend or stretch sensors. Vision-based methods have not been analyzed in this survey, because they are not really part of the gloves and do not support tracking beyond the field of view. Nevertheless, recently some smart gloves also include this technology and some kind of sensor fusion, such as SenseGlove Nova or VRFree. As a result, it is very difficult to provide a clear comparison. In this review, we have struggled to analyze the DoF capabilities, but other features such as motion range, sensing accuracy and update rate would be needed in order to have a clear picture. Information about these features is particularly difficult to find for the smart gloves and it is usually heterogeneous in nature, as some manufactures provide certain data, while other do not, which makes it difficult to compare. In any case, features obtained by vendors have been compiled in Table 9.

Many of the previous variability can be related to the desired application. IMU based gloves are usually found in motion capture initiatives where the key goal is to recognize the general position. Meanwhile, remote manipulation and videogames are more interested in capturing precise hand and finger movements to support the natural interaction of the user and provide a fully immersive and embodiment experience. By the contrary, medicine applications are usually related to the hand rehabilitation where the need for a precise tracking of the movements is not so important. Some other products cannot be clearly framed on some particular application area, but they are offered as a general-purpose solution, such as Hi5 VR or Manus. This review shows the existence of different approaches and it is not clear if in the future, a general-purpose solution will be available or application-specific products will be adopted.

Finally, there are a few other issues that have not been analyzed in this review but that would have a certain importance in order to select a smart glove. One issue is related to the calibration, particularly in the case of using an off-the-shelf IMU. Smart gloves have to be fitted to the human hand and fingers, which have a great variety of sizes and shapes depending on the person. Therefore, companies usually provide different gloves sizes. Nevertheless, because the sensor position along the hand and fingers is very important in order to detect the movements, a calibration process is needed before gloves can be used. Usually, companies include specific procedures and special software to support this in a more or less autonomous way. As an interesting example, Exo gloves offer a hand scanner that facilitates the measurement of the human hand and speeds up the calibration process. Related to the previous one, the reliability and lifespan of sensors is another main issue, particularly in the case of bend sensors, as they typically suffer from continuous operation. Another issue is related to the interoperability with XR platforms, such as programming frameworks (e.g., Unreal, Unity) and platforms (e.g., Oculus Rift, HTC Vive, Windows Mixed Reality). In this case, there exists a great variability of options that change frequently in short time. At this point, the development of open interoperable frameworks that facilitates the integration of third parties, particularly researchers, would be very interesting.

## 9. Conclusions

This review focused on active commercial smart gloves shows that there exists a great interest in these devices. There is a large number of running initiatives and, most importantly, many pieces of research are being developed involving commercial products, from sign language gesture detection to XR interaction. There are main differences in relation to previous years where the adoption were not so clear, as long as published reviews about smart gloves had not identified a so large number of solutions and applications in the commercial sector. We conclude that there is a recent trend that demonstrates and improved technological performance, particularly current devices are really portable and autonomous, and the maturity of the solutions captures real interest from users and researchers.

In any case, we would like to note that this review covers just a part of all the research on this field: active commercial smart gloves. Many research prototypes are also under development, involving the use of new types of sensors or materials. Similarly, vision-based methods have not been considered in this paper, but they are also being explored to recognize hand and finger position and tracking. Despite some problems that have not a clear solution, particularly occlusion, they also offer good performance in certain situations.

The first contribution of this paper is the precise description of the human and finger hand DoF and subsequent commercial smart gloves analysis. This provides a clear reference to compare the solutions available in the market. In addition, it also helps to understand the features and limitations of the various technologies. Another contribution is the review itself. To the best of our knowledge, this is the first time a review methodology, such as PRISMA, has been applied to perform a review about commercial smart gloves. This will facilitate the reproduction of the study in the future and the comparison with the current situation. From the review, we have identified two new glove types, strips of fabric and open fingertips, in addition to the generally recognized ones, exoskeleton and fabric. Similarly, smart gloves have been classified in accordance with three main capabilities: hand and finger pose estimation and motion tracking, kinesthetic feedback and haptic feedback. The review of the technologies shows a predominance for IMU and resistive bend sensors, but with a lot of variations and a trend towards the combination of technologies and sensor fusion. In addition, other related features are usually included: pressure sensors, screen interaction and touch points. All this will facilitate the comparison among smart gloves.

Finally, from our point of view, based on the depicted situation, it is very important to define frameworks that allow us to check and compare the different solutions available against clear references. The identification of DoF in commercial smart gloves is an example of the difficulties involved, not only because the partial and incomplete information provided by the vendors, but also because the differences among technologies. In this context some papers have been published comparing the hand pose recognition for specific gloves [26,34]. In any case, other features should be considered beyond, such as indicators to precisely compare the performance, and interoperability and integration of solutions, enabling users to move from the solutions of one vendor to another. There is a need for standardization and open-source initiatives that would contribute to a better development of this market, mainly for the development of final applications and for the application in research.

## Figures and Tables

**Figure 1 sensors-21-02667-f001:**
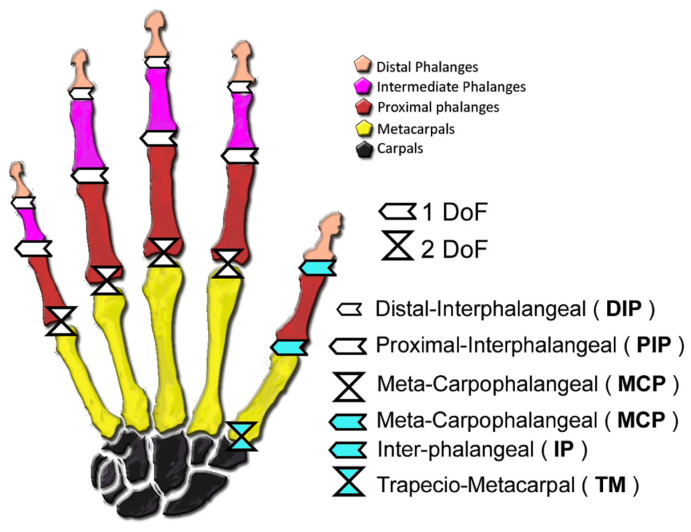
Human hand model.

**Figure 2 sensors-21-02667-f002:**
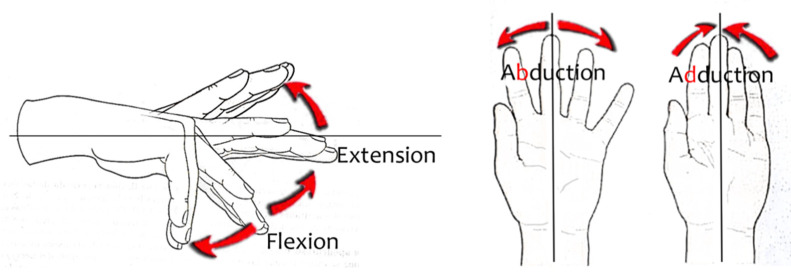
Finger movements (except thumb).

**Figure 3 sensors-21-02667-f003:**
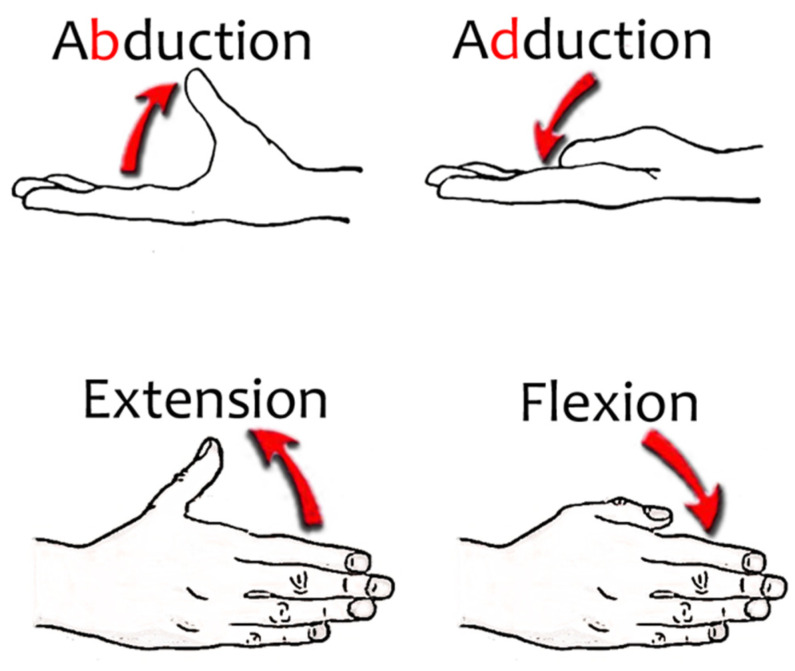
Thumb movements.

**Figure 4 sensors-21-02667-f004:**
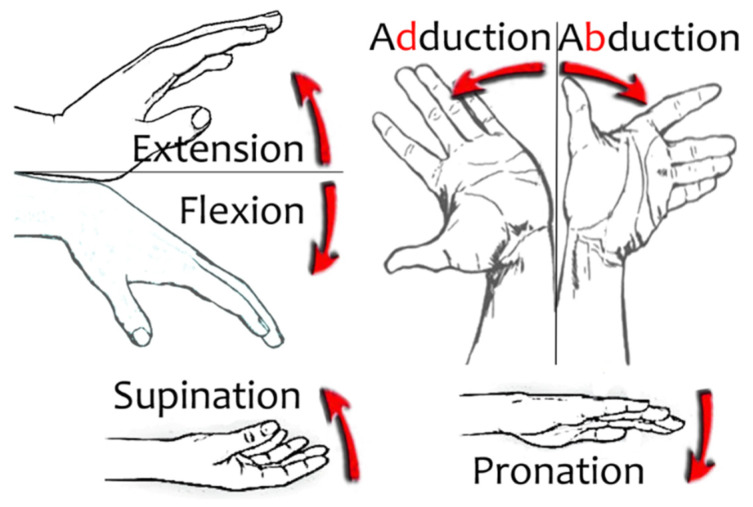
Wrist movements.

**Figure 5 sensors-21-02667-f005:**
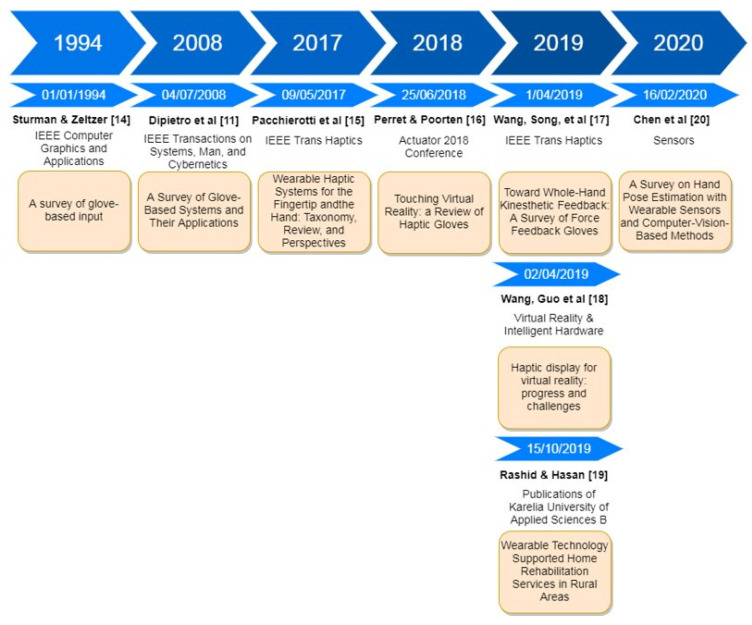
Published reviews about smart gloves timeline.

**Figure 6 sensors-21-02667-f006:**
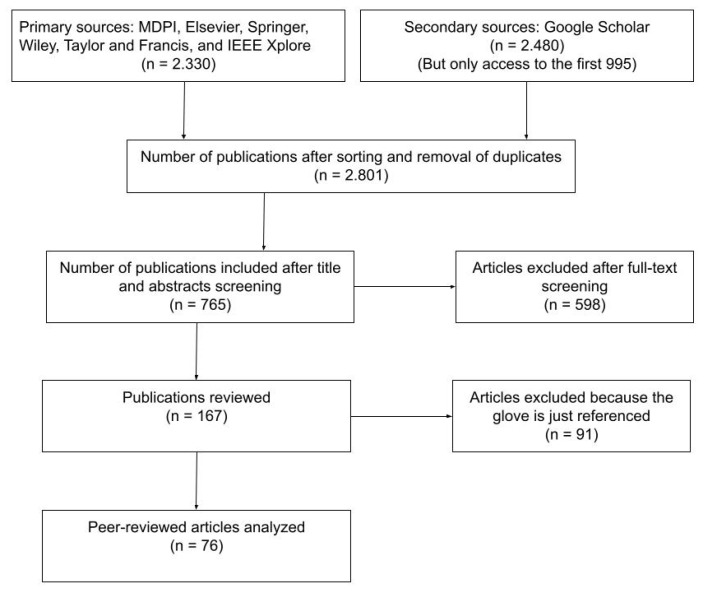
Main results of the PRISMA literature review stages.

**Figure 7 sensors-21-02667-f007:**
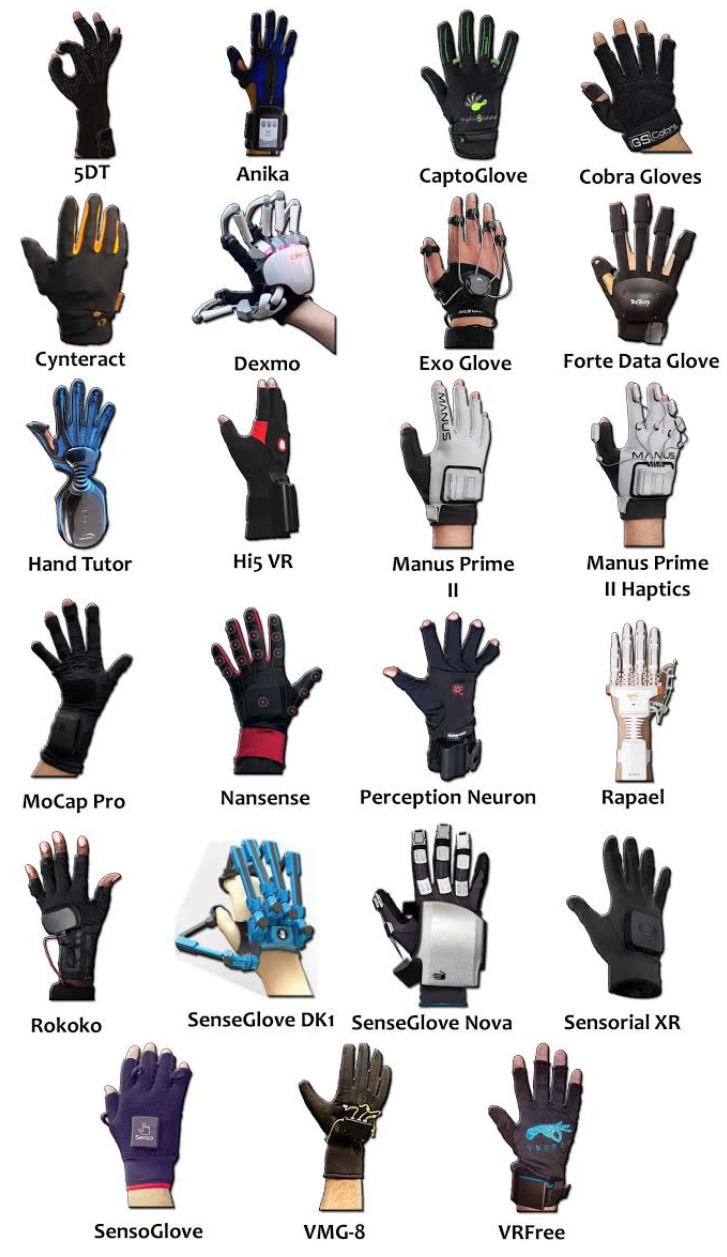
Pictures of commercial smart gloves (the VMG 35 Haptic is not included because it is like the VMG-8).

**Table 1 sensors-21-02667-t001:** Discarded gloves.

Smart Glove	Country	URL ^1^
AcceleGlove	USA	anthrotronix.com (accessed on 27 February 2021)
Avatar VR	Spain	neurodigital.es (accessed on 27 February 2021)
CyberGlove	USA	cyberglovesystems.com (accessed on 27 February 2021)
Cybertouch	USA	cyberglovesystems.com (accessed on 27 February 2021)
CyberGrasp	USA	cyberglovesystems.com (accessed on 27 February 2021)
CyberForce	USA	cyberglovesystems.com (accessed on 27 February 2021)
Esoglove	Singapore	roceso.com/esoglove-pro (accessed on 27 February 2021)
ExoHand	Global	festo.com/group/en/cms/10233.htm (accessed on 27 February 2021)
EXOS wrist DK2	Japan	exiii.jp (accessed on 27 February 2021)
FingerTracking	Germany	ar-tracking.com/en/product-program/fingertracking (accessed on 27 February 2021)
Glohera Sinfomia	Italy	gloreha.com/sinfonia/ (accessed on 27 February 2021)
GloveOne	Spain	neurodigital.es (accessed on 27 February 2021)
GoGlove	USA	goglove.io (accessed on 27 February 2021)
Haptic Pivot	Global	microsoft.com/en-us/research/publication/haptic-pivot-on-demand-handhelds-in-vr (accessed on 27 February 2021)
Haptx	USA	haptx.com (accessed on 27 February 2021)
HGlove	France	haption.com/en/products-en/hglove-en.html (accessed on 27 February 2021)
HumanGlove	Italy	hmw.it/en (accessed on 27 February 2021)
KeyGlove	USA	keyglove.net (accessed on 27 February 2021)
Maestro Gesture Glove	Canada	maestroglove.com (accessed on 27 February 2021)
Music Glove	USA	flintrehab.com/product/musicglove-hand-therapy (accessed on 27 February 2021)
Nuglove	USA	anthrotronix.com (accessed on 27 February 2021)
Peregrine	Canada	peregrineglove.com (accessed on 27 February 2021)
Plexus	UK	plexus.in (accessed on 27 February 2021)
Polehemus	Canada	polhemus.com/motion-tracking/hand-and-finger-trackers (accessed on 27 February 2021)
SaeboGlove	USA	saebo.com/shop/saeboglove (accessed on 27 February 2021)
Tactile Glove	Global	pressureprofile.com/body-pressure-mapping/tactile-glove (accessed on 27 February 2021)
TeslaSuit Gloves	USA	teslasuit.io/blog/vr-glove-by-teslasuit/ (accessed on 27 February 2021)
Valve Index Controllers	USA	valvesoftware.com/es/index/controllers (accessed on 27 February 2021)
VRGluv	USA	vrgluv.com/enterprise (accessed on 27 February 2021)

^1^ All URLs were last accessed at 27 February 2021.

**Table 2 sensors-21-02667-t002:** Commercial smart gloves (first row abbreviations: Tr. = “Hand and finger pose estimation and motion tracking”; KF = “Kinesthetic Feedback”; TF = ”Tactile Feedback”; first column abbreviations: v. = “versions”).

Smart Glove	Country	URL ^1^	Glove Type	Tr.	KF	TF	Price
5DT (2 v.)	USA	5dt.com	Open tips	X			$2990–$5495
Anika Rehap	Russia	zarya-med.com	Strips	X			$1300
CaptoGlove	Italy	captoglove.com	Fabric	X			$315
Cobra Glove	Germany	synertial.com	Open tips	X			$7450
Cynteract	Germany	cynteract.com/en	Fabric	X		X	€500
Dexmo (3 v.)	China	dextarobotics.com	Exoskeleton	X	X	X	$36,000
Exo Glove (3 v.)	Germany	synertial.com	Strips ^2^	X			$4980
Forte Data Glove	USA	bebopsensors.com	Strips	X		X	$3000
HandTutor	Israel	handtutor.com	Fabric	X			€3400
Hi5 VR	China	noitom.com	Open tips	X		X	$999
Manus Prime II	The Netherlands	manus-vr.com	Open tips	X			€1499
Manus Prime II Haptics	The Netherlands	manus-vr.com	Open tips	X		X	€2499
MoCap Pro SuperSplay	New Zealand	stretchsense.com/product/mocap-pro-super-splay	Fabric	X			$7150
Nansense R2 (3 v.)	USA	nansense.com/	Fabric	X			$4798
Perception Neuron Studio Gloves	China	neuronmocap.com/content/product/perception-neuron-studio-gloves	Open tips	X			$1499
Rapael	Germany	neofect.com	Strips	X			$1925
Rokoko	Danish	rokoko.com	Open tips	X			$995
SenseGlove DK1	The Netherlands	senseglove.com	Exoskeleton	X	X	X	€2999
SenseGlove Nova	The Netherlands	senseglove.com	Exoskeleton ^3^	X	X	X	€4500
Sensorial XR	Spain	neurodigital.es	Fabric	X		X	€11,995
Senso Glove DK3	USA	senso.me/order	Open tips	X		X	$999
VMG (4 v.)	USA	virtualmotionlabs.com	Fabric	X			$1000-
VMG 35 Haptic	USA	virtualmotionlabs.com	Fabric	X		X	-
VRFree ^4^	Switzerland	sensoryx.com	Open tips	X			CHF750

^1^ All URLs were last accessed on 27 February 2021. ^2^ Includes rings for each of the fingers and thumb. ^3^ The SenseGlove Nova is actually a kind of tighten exoskeleton, wrapped around the hand, similar to a strips glove. ^4^ VRFree also offers a Haptic version but no information was found about features.

**Table 3 sensors-21-02667-t003:** Ergonomics and wearability features in commercial smart gloves.

Smart Glove	Size	Weight (Grams)	Battery and Autonomy	Connection	Other Features
5DT	Unique	NA	A battery pack	USB, RS232 Bluetooth	- Black stretch Lycra - Wireless kit
Anika Rehap	Adaptable	200	Wired provided	USB	
CaptoGlove	Unique	NA	Li-Ion Polymer 10 h	BLE 4.0	- Fabric glove - Washable and breathable
Cobra Glove	4 (SS, S, M, L)	70–150	AA batteries	Wi-Fi	- Detachable electronics
Cynteract	3 (S, M, L)	NA	NA	USB	
Dexmo	Unique	300	Li-Ion Polymer 5 h	Wi-Fi 2.4 USB	- Memory foam hand pad - Mechanism to minimize sweating
Exo Glove	3 (S, M, L)	145	External AA batteries	Wi-Fi 2.4 BLE 5.0, SD Card	- Modularity - Finger freedom with ring system
Forte Data Glove	Unique	103.5	Li Polymer 6–8 h	BLE USB	- Neoprene, nylon and Lycra
HandTutor	5	200	No	USB	
Hi5 VR	2 (S, M)	105	AA batteries 3 h	Wi-Fi 2.4	- Antibacterial, breathable elastic textile
Manus Prime II ^1^	Unique	60	Batteries 5 h	Wi-Fi 2.4	- Antibacterial - Sports polyester
MoCap Pro SuperSplay	2 (S/M, M/L)	110	Battery 8 h	Bluetooth, Wi-Fi, USB-c, SD card	- Antibacterial, breathable, stretchy fabric - Palm rubberized grips - Velcro for optical markers
Nansense R2	3 (S, M, L)	255	Battery 6-8 h	Wi-Fi 2.4, 5 USB-A	- Single piece of fabric - Velcro for markers - No calibration automatic sensor compensation
Perception Neuron	3 (S, M, L)	105	AA batteries 5 h	Wi-Fi	
Rapael	Unique	132	Battery	Bluetooth	- Elastomer material - Easy cleaning
Rokoko	4 (S, M, L, XL)	70	External power bank	Wi-Fi 2.4 Wi-Fi 5	- Tight fit to keep sensor in place
SenseGlove DK1	Unique	300	Li-Ion battery 2 h	USB Bluetooth	- Plastic and fabric - Wireless kit
SenseGlove Nova	Unique	315	Battery 4 h	Bluetooth	- Kind of armored glove
SensorialXR	Unique	140	600 mA Li-Po 6-8 h	BLE 5.0	- Lycra with antibacterial - Fire-resistant treatments
Senso Glove DK3	5 (S, M, ML, L, XL)	300	Li-Ion Polymer 1.5 h	USB RF, BLE	
VMG ^1^	NA	NA	Li-Po Battery 5–6 h	USB, Bluetooth	
VRFree	4 (S, M, L, XL)	40	Replaceable rechargeable batt.	Wireless USB-C	- Multiple sensor types - A module has to be clipped on an HMD headset.

^1^ All versions share the same features.

**Table 4 sensors-21-02667-t004:** Commercial smart gloves for hand and finger pose estimation and motion tracking.

Smart Glove	Sensor Technology	IMU	Sensors	External IMU
5DT (2 v.)	Fiber optic bend sensors	No	5/14	No
CaptoGlove	Bend sensors	1	5	No
Cobra Glove (3 v.)	IMU	7/13/16	No	Yes
Dexmo (3 v.)	Mechanical rotational sensors	No	5	Yes
Exo Glove	IMU	6	No	Yes
Forte Data Glove	Bend sensors and IMU	1	10	Yes
HandTutor	Bend sensors	No	5	No
Hi5 VR	IMU/Optical hybrid	6	No	Yes
Manus Prime II	Resistive bend sensors and IMU	1	10	Yes
M. Prime II Haptics	Resistive bend sensors and IMU	1	10	Yes
MoCap Pro SuperSplay	3 sensing zones splay sensors	No	6	Yes
Nansense R2 (3v.)	IMU	7/12/15	No	Yes
Perception Neuron	IMU	6	No	No
Rapael	Resistor bend sensors and IMU	1	5	No
Rokoko	IMU without magnetometers	7	No	No
SenseGlove DK1	IMU and rotation encoders	1	20	Yes
SenseGlove Nova	IMU + Vision (Pico Neo 2)	1	5	Yes
SensorialXR	IMU	7	No	No
Senso Glove DK3	IMU	8	No	No
VMG (4 v.)	IMU and bend sensors	5/5/16/0	1	No
VMG 35 Haptic	IMU and bend sensors	21	1	No
VRFree ^1^	6 sensor types: bend, IMU, etc.	NA	NA	No

^1^ VRFree refers the use of 6 different, complementary sensor types that are fully integrated, but without detailing neither types, number nor location.

**Table 5 sensors-21-02667-t005:** DoF for commercial smart gloves (av. is used to indicate that the average movement of two joints is measured).

Smart Glove	DoF	Four Fingers	Thumb	Other Ones
5DT 5 sensors	5 ^1^	E/F (av. PIP, MCP)	E/F (av. PIP, MCP)	No
5DT 14 sensors	14 ^1^	E/F × 2(PIP, MCP) + A/A	E/F × 2 (PIP, MCP)	No
CaptoGlove	5 ^2^	E/F (av. DIP, PIP)	E/F × 2 (av. PIP, MCP)	No
Cobra Glove 7	7 ^3^	E/F (PIP)	E/F × 2 (PIP, MCP)	Palm ^5^
Cobra Glove 13	13 ^3^	E/F × 2 (PIP, MCP)	E/F × 3 + A/A	Palm ^5^ × 2
Cobra Glove 16	16 ^3^	E/F × 3 − 2 ^4^	E/F × 3 + A/A	Palm ^5^ × 2
Dexmo	10	E/F (MCP) + A/A	E/F (MCP) + A/A	No
Exo Glove	6 ^3^	E/F (av. PIP, MCP)	E/F × 2 ((av. PIP, MCP), TM)	No
Forte Data Glove	13 ^1,6^	E/F + A/A	E/F + A/A	Wrist E/F + A/A + P/S
HandTutor	5 ^1^	E/F (av. DIP, PIP, MCP)	E/F (av. PIP, MCP)	No
Hi5 VR	8 ^3^	E/F	E/F	Wrist E/F + A/A + P/S
Manus Prime II	14 ^1^	E/F × 2	E/F × 2 + A/A	Wrist E/F + A/A + P/S
Manus Prime II Haptics	14 ^1^	E/F × 2	E/F × 2 + A/A	Wrist E/F + A/A + P/S
MoCap Pro SuperSplay	11	E/F + AA	E/F + A/A	Wrist A/A
Nansense R2 7	7 ^3^	E/F (PIP)	E/F × 2 + A/A	Palm ^5^
Nansense R2 12	12 ^3^	E/F × 2 (PIP, MCP)	E/F × 3 + A/A	Palm ^5^
Nansense R2 15	15 ^3^	E/F × 3 − 2 ^4^	E/F × 3 + A/A	Palm ^5^
Perception Neuron	8 ^3^	E/F	E/F	Wrist E/F + A/A + P/S
Rapael	5	E/F (av. DIP, PIP, MCP)	E/F (av. PIP, MCP)	Wrist E/F + AA + P/S
Rokoko	9 ^3^	E/F	E/F + A/A	Wrist E/F + AA + P/S
SenseGlove DK1	23 ^7^	E/F × 3 + A/A	E/F × 3 + A/A	Wrist E/F + A/A + P/S
SenseGlove Nova	8 ^8^	E/F	E/F + A/A	Wrist E/F + A/A + P/S
SensorialXR	9 ^3^	E/F	E/F + A/A	Wrist E/F + A/A + P/S -
Senso Glove DK3	9 ^3^	E/F	E/F + A/A	Wrist E/F + A/A + P/S
VMG 8	8 ^1^	E/F	E/F	Wrist E/F + A/A + P/S
VMG 13	8 ^1^	E/F	E/F	Wrist E/F + A/A + P/S
VMG 30	19 ^1^	E/F × 2 + A/A	E/F × 2 + A/A	Wrist E/F + A/A + P/S + Palm ^5^
VMG 35 Haptic	23 ^1^	E/F × 3 + A/A	E/F × 3 + A/A	Wrist E/F + A/A + P/S + Palm ^5^
VMG PS	3 ^1^	No	No	Wrist E/F + A/A + P/S
VRFree	23 ^9^	E/F × 3 + A/A	E/F × 3 + A/A	Wrist E/F + A/A + P/S

^1^ Estimated, taking into account the number of sensors and their location. ^2^ The CaptoGlove includes five sensors situated over the fingers (PID and DIP joins) and the vendor indicates that it provides 10 DoF, with two DoF per finger. This is a bit strange. Probably, each sensor is used to provide the average value of the two DoF. ^3^ Estimated, based on IMU. In case of Cobra Glove versions with seven and 13 sensors, interpolation to approximate the untracked finger joints is used. In the case of Manus Prime II an IMU sensor is located at the thumb MCP. ^4^ All fingers have three E/F movements, except for the pinky fingertip that just has two E/F movements (the DIP is not included). ^5^ Palm bending is detected. ^6^ The Forte Data Glove vendor indicates 28 DoF. According to our model, this is beyond the possible DoF for the human hand. ^7^ The SenseGlove DK1 has 23 DoF that have been described in [34]. ^8^ The SenseGlove Nova eight DoF are estimated considering just the sensors available. The vendor indicates additional DoF can be obtained by fusion with proprietary vision-based algorithms. ^9^ The VRFree indicates 23 DoF are obtained by fusion of different sensor technologies, including vision-based algorithms.

**Table 6 sensors-21-02667-t006:** Commercial smart gloves for kinesthetic feedback.

Smart Glove	Mode	Technology	Actuated DoF	Force
Dexmo	Active	Servo motors	5	0.3 N m
SenseGlove DK1	Passive	Magnetic friction brakes and strings	5	40 N
SenseGlove Nova	Passive	Brakes and mechanical wires	4 ^1^	20 N

^1^ From thumb to ring finger.

**Table 7 sensors-21-02667-t007:** Commercial smart gloves for tactile feedback.

Smart Glove	Technology	Actuators Number	Location	Type
Dexmo	LRA	6	Fingertips, thumb and palm	Programmable
Forte Data Glove	Non-resonant actuators	6	Fingertips, thumb and palm	Programmable
Hi5 VR	Vibration “rumbler”	1	Wrist	Programmable
Manus Prime II Haptics	LRA	5	Fingertips and thumb	Programmable
SenseGlove DK1	Vibration motors	6	Fingertips, thumb and palm	Collisions, textures, button clicks
SenseGlove Nova	2 LRA 1 Voice coil	3	Thumb, index finger and hand ^1^	Fell shapes, textures, stiffness, impacts and button clicks
SensorialXR	Customized low-latency LRA	10	Palm, thumb, index and middle finger	1024 vibration profiles
Senso Glove DK3	LRA vibration motor	6	Fingers and thumb (under the last phalange)	More than 100 haptic effects
VMG 35 Haptic	Vibro-tactile actuators	5	Fingers and thumb	Programmable

^1^ Thumb and index finger have a vibro-tactile actuator each one. The voice coil is located at the hub of the glove.

**Table 8 sensors-21-02667-t008:** Commercial smart gloves by application area.

Smart Glove	Medicine & Remote Healthcare	Motion Capture	Video Game	Simulation & Training	Manipulation of 3D Objects	XR Applications
5DT	[35,36,37]	[36,38,39,40,41]		[42]		
Anika Rehap	X					
CaptoGlove	X		X	X	X	X
Cobra Gloves	X	[43]		X	X	
Cynteract	X		X		X	
Dexmo	X	[44]	X	X	[45,46]	
Exo Gloves		X		X	X	
Forte Data Glove	X			[47]	X	X
HandTutor	[48]					
Hi5 VR	[49,50,51]	[52]	X	[53,54]	[55,56,57,58,59]	[60,61]
Manus	[62]	[63]	[64]	[65]	[63,66,67,68]	[69]
MoCap Pro S.		X				
Nansense R2	X	X	X			[70]
Perception Neuron	[71]	[72]	X			X
Rapael	[73,74,75,76,77,78,79]					
Rokoko		X				
SenseGlove DK1	X			[80]	[81,82,83]	[84,85]
Sense Glove Nova					X	
Senso Glove	X	[86]	X	[87,88]		[89]
SensorialXR	X			X	X	X
VMG	[90]	[91,92,93]	[94]	X	X	X
VRfree		X	X	X	[34]	X

**Table 9 sensors-21-02667-t009:** Commercial smart gloves metrological features.

Smart Glove	Features
5DT	- Continuous data for each sensor: 0–1 - Minimum sampling rate for the full hand (all available sensors): 75 Hz - Flexure resolutions: 12-bit A/D for each sensor - Minimum dynamic range: 8 bits (256 angular values) per joint
CaptoGlove	- Extension/flexion movements resolution: <1 degree - Tactile sensor: 1 pressure sensor for thumb’s fingertip 100 g–10 kg
Cobra Glove	- Internal update rate: 500 Hz - Gyro range: 2000 degrees/s - Accelerometer range: 1–6 Gs
Dexmo	- Kinesthetic-feedback: 1 DoF per finger, with a maximum force of 0.3 Nm - Frequency Transmission Range: 2.4 GHz - Accuracy: +/−0.5 degrees
Exo Glove	- Gyro rotation vector: 1000 times/s - Rotation vector: 400 times/s - Gravity: 400 times/s - Linear acceleration: 400 times/s - Accelerometer: 500 times/s - Gyroscope: 400 times/s - Magnetometer: 100 times/s
Forte Data Glove	- Accuracy and repeatability: +/−1.5 degrees - Latency: 150 frames/sec (<6 ms) - Sensor performance sample rate: 200 Hz - Frequency response: 100–2000 Hz
HandTutor	- Sensitivity: 0.05 mm of wrist and fingers - Motion capture speed: Up to 1 m/s
Hi5 VR	- Latency: <5 ms - Data rate: Up to 180 Hz
Manus Prime II ^1^	- Sensor sampling rate: 90 Hz - Orientation accuracy: +/−2.5 degrees - Signal latency: <5 ms - Finger flexible sensor repeatability: >1,000,000 cycles - Orientation sensor accuracy: +/−2.5 degrees
Nansense R2	- Data rate: 240 fps - Latency: +/−30 ms
Perception Neuron	- Accelerometer range: +/−8 g - Gyroscope range: +/−2000 dps - Resolution: 0.02 degree - Frequency: 2400–2483 MHz - Accuracy: Roll 0.7°/Pitch 0.7°/Yaw 2° - Internal processing rate: 800Hz - Output rate: 60/90/96/100 Hz
Rokoko	- Frequency: 400 Hz - 3D orientation accuracy: +/−1 degree - Data rate: 100 fps - Latency: +/−20 ms
SenseGlove DK1	- Force feedback output: 40 N
Senso Glove DK3	- Frequency: 400 Hz - Latency: 15 ms
VMG ^1^	- Finger Sensing Resolution: 12 bit (4096 step) - Sampling rate: 10–100 Hz - Accuracy: Roll +/−0.01 °/Pitch +/−0.01°/Yaw +/−0.05° - Scale range: +/−2 g, +/−4 g, +/−8 g
VRFree	- Data rate: 100 MHz - Orientation resolution: 0.01 degree - Displacement resolution: 0.3 mm - Frequency: 120 Hz (8 ms)

^1^ All versions share the same features.

## Data Availability

No new data were created or analyzed in this study. Data sharing is not applicable to this article.

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
