# Peer review of "A Systematic Review of Commercial Smart Gloves: Current Status and Applications"

_sensors, 2021, doi:10.3390/s21082667_

Round 1

Reviewer 1 Report

The paper provides a systematic review of currently available commercial devices, namely smart gloves, providing (i) hand and finger pose estimation and motion tracking, (ii) kinesthetic feedback, and (iii) tactile feedback for human-computer interaction.

The authors conduct a well-documented systematic literature review using the PRISMA guidelines resulting in 24 currently commercially available devices.  The functionality of the considered devices regarding  (i) hand and finger pose estimation and motion tracking, (ii) kinesthetic feedback, and (iii) tactile feedback for human-computer interaction is presented according to their use or analysis that is carried out on the reviewed papers.

The paper is well organized and only minor issues could be suggested for improvements which are to be found in the next paragraphs.

  1. Regarding the PRISMA guidelines, a suggestion to the authors is to change the title of their paper in order to be in compliance with PRISMA by identifying their report as a "systematic review".
  2. The abstract is structured and well written, nevertheless, the suggested structure with headings as recommended by PRISMA is not followed.
  3. A statement on whether a protocol is registered and used is missing (PRISMA guidelines).
  4. Except for the previously given minor issues regarding PRISMA, an interesting addition, although authors state its absence, is the interoperability with programming frameworks and VR/XR platforms. It is understandable that such an analysis may be beyond the scope of this paper, but if the considered information could be extracted from the reviewed papers and briefly presented it would be of great interest to the readers.

The paper is well written, with a significant contribution to the field and of potential interest to the readers. 

Author Response

Thank you very much for your comments. Regarding the PRISMA methodology, we would like to say that we carry out the study based on PRISMA, but as this work is performed in a technological context and not in the health care one as it was originally conceived, we don’t follow all the PRISMA indications. Please, find the answer to each one of the minor issues suggested:

  1. As suggested by the reviewer we have changed the paper title identifying the report as a "systematic review.
  2. We understand the suggestion of the review, but as the Sensors journal has specific instructions for the abstract, we cannot approach this request. Particularly, the number of works is limited to 200 and there is an indication of not using headings. The provided abstract is similar to existing abstracts in recent reviews published by Sensors.
  3. Due to PRISMA is mainly oriented to healthcare, and to the best of our knowledge, this is the first study in this field following the PRISMA methodology, we did not consider necessary to record the revision protocol. In any case, we have included a sentence indicating this in section 4.
  4. As the reviewer indicates we consider this issue is beyond the scope of this paper. We had already included some comments in section 8, lines 869-87 (in the new version 878-882). A main issue is that interoperability with XR platforms and programming frameworks is continuously changing as vendors update their support quite dynamically.

Reviewer 2 Report

This paper is a review of the current commercial smart gloves, able to estimate hand and finger pose or track motion as well as to provide kinesthetic feedback or  tactile feedback.. Based on the PRISMA guidelines for systematic reviews for the period 2015-2021, the author identified 24 commercial smart glove and they were classified according to the type, their capabilities, the ergonomics and wearability features (size, weight, type of sensors, connection type).

The paper is very interesting and strong related to the journal topics. The review of the state of the art is appropriately elaborated. Methodology is well introduced. 

The authors provide very helpful information for the general audience. In order to provide a significant contribution to the reaserch, more metrological information could be added, such as accuracy, resolution or stability. This information can be found either from the manufacturers' specifications or from the papers that the authors have analyzed. If it is not possible to find this information for all the devices, it may be sufficient to provide just a range. Finally, it might be interesting to mention works and their results comparing the performance of some of the analysed devices.
All this information could be useful to identify the typical issues fo the smart gloves (some of them have already been identified) and help the researchers understanding which issues need to be solved (ergonomics, sensors, algorithms, etc.).

Author Response

Thank you for your comments. We consider these suggestions are of interest and we have updated the paper.

Regarding the metrological information we have included the available data in a new table in section 8. As we have indicated, this information is quite heterogeneous.

Regarding works comparing the performance of some of the analyzed devices we have identified two in the literature. We have included a comment in the conclusions section, referencing these two works.